# The first Australian plant foods at Madjedbebe, 65,000–53,000 years ago

S. Anna Florin [1✉], Andrew S. Fairbairn[1,2,3], May Nango[4], Djaykuk Djandjomerr[4], Ben Marwick[5], Richard Fullagar [6], Mike Smith [7,8], Lynley A. Wallis[9,10] & Chris Clarkson [1,2,3✉]

There is little evidence for the role of plant foods in the dispersal of early modern humans into new habitats globally. Researchers have hypothesised that early movements of human populations through Island Southeast Asia and into Sahul were driven by the lure of high-calorie, low-handling-cost foods, and that the use of plant foods requiring processing was not common in Sahul until the Holocene. Here we present the analysis of charred plant food remains from Madjedbebe rockshelter in northern Australia, dated to between 65 kya and 53 kya. We demonstrate that Australia's earliest known human population exploited a range of plant foods, including those requiring processing. Our finds predate existing evidence for such subsistence practices in Sahul by at least 23ky. These results suggest that dietary breadth underpinned the success of early modern human populations in this region, with the expenditure of labour on the processing of plants guaranteeing reliable access to nutrients in new environments.

[1] School of Social Science, University of Queensland, Brisbane, QLD 4072, Australia. [2] Australian Research Council Centre of Excellence for Australian Biodiversity and Heritage, University of Wollongong, Wollongong, NSW 2522, Australia. [3] Depatrment of Archaeology, Max Planck Institute for the Science of Human History, Kahlaiche Strasse 10, 07745 Jena, Germany. [4] Gundjeihmi Aboriginal Corporation, 5 Gregory Place, Jabiru, NT 0886, Australia. [5] Department of Anthropology, University of Washington, Seattle, WA 98195, USA. [6] Centre for Archaeological Science, School of Earth, Atmospheric and Life Sciences, University of Wollongong, Wollongong, NSW 2522, Australia. [7] College of Humanities, Arts and Social Sciences, Flinders University, Adelaide, SA 5042, Australia. [8] Centre for Historical Research, National Museum of Australia, Canberra, ACT 2601, Australia. [9] Nulungu Research Institute, University of Notre Dame Australia, Broome, WA 6725, Australia. [10] Present address: Griffith Centre for Social and Cultural Research, Griffith University, Brisbane, QLD 4111, Australia. ✉email: stephanie.florin@uqconnect.edu.au; c.clarkson@uq.edu.au

The role of plant foods in the evolution and dispersal of early modern humans (EMHs) has often been underestimated. A long-held focus on the notion of Paleolithic populations as meat eaters and a lack of consistent archaeobotanical recovery has frequently constrained analysis and understandings of EMH diet to its animal components[1]. Extensive use and processing of plant resources, and an associated broadening of the diet, was therefore typically considered a late Pleistocene/early Holocene phenomenon, linked to changing foraging behaviours in the millennia prior to the emergence of agriculture[2,3]. However, while plant foods may not make up the dominant proportion of EMH diets globally, more recent research into plant macro- and micro-fossils is breaking down this paradigm: the use of plant foods, including those associated with later agricultural transitions, such as grass seeds and underground storage organs (USOs), is now evidenced in Middle Stone Age sites in Africa and the Middle East[4–7]; the processing of toxic plants (*Dioscorea hispida* and *Pangium edule*) is now dated to as early as 46–34 kya in Niah Cave, Borneo[8–10]; the translocation of yams (*Dioscorea* spp.) to high altitudes and management of monodrupe pandanus stands, facilitated early use of highland environments in New Guinea (~49 kya)[11,12]; and associated plant-processing technologies, such as seed-grinding stones, are linked to EMH dispersal into northern Australia[13].

This shift in paradigm is particularly important when considering the southern dispersal of *Homo sapiens* out of Africa. Key debates in this region have focused on both the 'modernity' of EMH populations involved in these migrations[14,15] and the 'pathways' they may have followed[16]. Proponents of the single coastal dispersal model suggest human populations expanded along coastal environments ~60 kya, moving quickly through Sunda and Wallacea, and into Sahul[17,18]. This model emphasises the lure of high-ranked coastal resources and suggests diet breadth was likely narrow in the earliest phases of human expansion throughout this region. In contrast, other models highlight early adaptations to non-coastal environments by EMHs leaving Africa[16,19,20], including to more extreme ecosystems (e.g. rainforests[8–10,21,22], high-altitudes[11,12] and deserts[23,24]). Current evidence for Pleistocene plant use in Sunda and Sahul, while not necessarily related to the earliest phases of human expansion in this region[18], largely supports this latter interpretation. This is because the intensive and multi-step processing techniques required to make the identified plant foods edible are indicative of both complex and flexible foraging, and the kind of broad diet that underpins adaptations to more difficult environments[1,25].

Here we report on the charred plant macrofossils recovered from the earliest layer of dense occupation at Madjedbebe. Madjedbebe is a rockshelter in western Arnhem Land (northern Australia) situated at the base of the Djuwamba Massif, an escarpment outlier to the east of the Magela Creek floodplain (Fig. 1). Its earliest, dense phase of occupation (Phase 2) contains charcoal, abundant ground ochre, grinding stones, including those used for seed-grinding, and a dense assemblage of unique flaked stone artefact types and raw materials (>10,000 artefacts). This phase is dated to c.65–53 kya on the basis of an extensive single-grain optically stimulated luminescence and

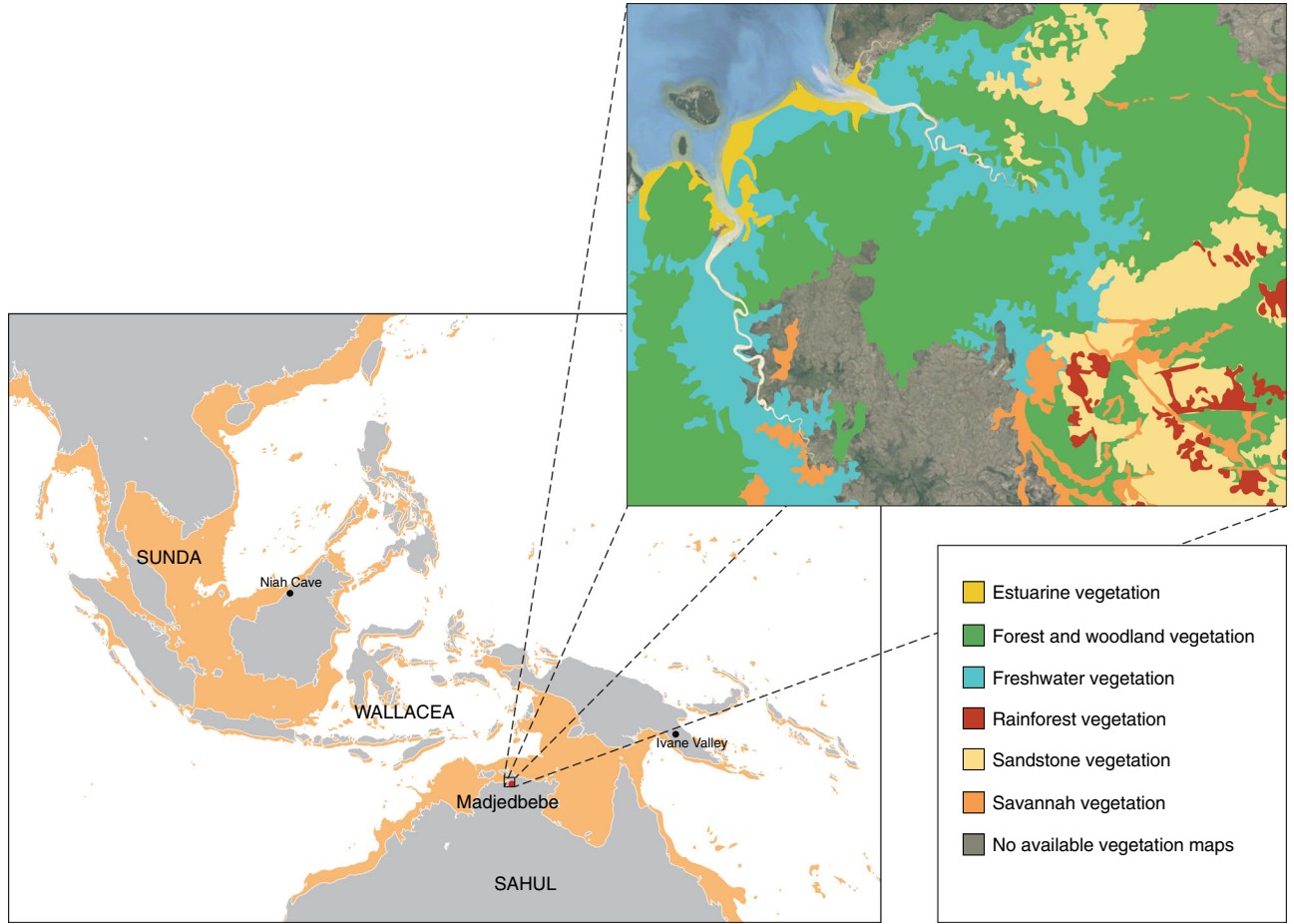

**Fig. 1 Site location. a** Regional map showing the location of Madjedbebe, and other sites mentioned in-text, adapted from Norman et al. 2018[47]; **b** current distribution of vegetation communities in proximity to Madjedbebe, developed from map data provided by Google, Landsat/Copernicus, TerraMetrics, and vegetation maps of the Adelaide and Alligator Rivers area[48,49].

accelerator mass spectrometry radiocarbon dating regime (Supplementary Fig. 1)[13,26,27]. The archaeobotanical assemblage from Phase 2, therefore, provides the earliest known evidence for an EMH diet in Sahul. We demonstrate that Australia's earliest known human population exploited a range of plant foods, including some requiring processing. Our findings have implications for understanding EMH behavior, cognitive flexibility and subsistence strategies at the eastern end of the modern human dispersal arc.

## Results

**Phase 2 plant macrofossil assemblage**. Plant macrofossil remains were recovered from all phases of occupation at the site using flotation (see Methods). The Phase 2 assemblage includes over 1000 non-wood plant macrofossils from a distinct hearth feature (C1/43 A) and excavated sediment matrix (see Supplementary Table 1). These macrofossils can be broadly categorised into four distinct groups: (i) endocarp and mesocarp ('nutshell' or 'fruit-stone') from a variety of fruit and nut producing species; (ii) vegetative parenchyma from USOs ('roots and tubers'); (iii) stem tissue from the Arecaceae (palm) family and (iv) various other fragments of plant material. Within these groups, genus-, or species-specific identifications have been made for five taxa, with a further five types identified to broader botanical categories.

The paucity of charcoal prior to human occupation (Supplementary Fig. 2a) and the presence of a diversity of edible species preserved by burning suggests that the assemblage was largely derived from human activities, specifically the cooking and disposal of plant resources in hearths. This is corroborated by the relatively high percentage (17%) of fragile parenchymatous tissue from USOs preserved in the assemblage. USO-producing species have evolved to survive and profit from bushfires, rapidly regenerating new aerial shoots from their buried vegetative organs[28]. These organs, themselves, are therefore unlikely to have been charred within the site without human activity. However, this does not preclude the inclusion of some of the Phase 2 plant macrofossils via non-human agents.

**Endocarp and mesocarp**. Charred endocarps of five fruit and nut taxa were identified: *Buchanania* sp.; *Canarium australianum*; polydrupe *Pandanus* sp.; *Persoonia falcata*; and *Terminalia* sp. (Fig. 2). All of these taxa are common in open forest and woodland, and/or monsoon vine forest environments[29], and the majority require little or no processing. Today, the abundant and easily harvested 'plums' of *Buchanania* spp. and *Persoonia falcata* are highly sought after. The fruits can be eaten raw but are also often ground into a paste, incorporating the endocarp and seed, prior to consumption (MN, DjDj). *Canarium australianum* is a relative of the widely used Melanesian tree-crop, galip (*C. indicum*), and has a small oil-rich kernel (<1 cm) that is easily extracted with a single blow from a hammerstone (DjDj). A number of *Terminalia* species are edible, consumed either as fruits (*T. carpentariae, T. erythrocarpa, T. ferdinandiana, T. microcarpa*) or easily-extracted nuts (*T. grandiflora*; MN, DjDj)[30]. However, there are also a number of non-edible species of *Terminalia* in the Northern Territory.

Pandanus formed a substantial element of the early colonial diet of Indigenous Australians in Arnhem Land and is also a valuable material in weaving and fibrecraft[31]. There are two types of polydrupe pandanus in Arnhem Land: *P. spiralis* and *P. basedowii*[29]. Only one fragment from Phase 2 at Madjedbebe, a portion of mesocarp, can be securely identified as *P. spiralis* (Fig. 3f). However, it is likely that the majority come from this species, since *P. basedowii* only grow on the escarpment top and would have been difficult to access from Madjedbebe (see Fig. 1b). Extracting kernels from the fibrous and mechanically-resistant prismatic structure of

the *P. spiralis* drupe is a labour-intensive process when using stone tools. The explorer, Ludwig Leichardt, recorded Indigenous groups in the Gulf of Carpentaria in the 1840s using "large flat stones and pebbles" to bash apart the drupes[32]. However, once open, the small kernels are rich in fat (44–50%) and protein (20–34%)[33].

**Vegetative parenchyma**. Three distinct types of vegetative parenchyma are present in the Phase 2 assemblage. These include parenchymatous tissue from two types of monocotyledonous, stem-based storage organs and a fragment from a secondary-root storage organ (Fig. 3). Monocotyledonous stem-based storage organ Type A is represented in Phase 2 only by charred fragments of its skin tissue, which have distinct root abscission scars, or 'eyes', and surface patterning (Fig. 3a, b). These fragments are comparable to charred peelings generated by contemporary Indigenous Australians when they remove the coarse external surface of cooked USOs before consumption, often in proximity to the hearth in which they were cooked (MN, DjDj; Fig. 3d). The presence of an endodermis, in a larger fragment of this type recovered from Phase 3, allows for its further identification as an aquatic or semi-aquatic USO (Fig. 3c). Endodermis is rarely present in stem tissue, except in aquatic plants where they are significant in controlling water balance within perennating stems[34].

**Arecaceae**. There are two types of Arecaceae stem tissue present in the Phase 2 assemblage (Fig. 4). Type A, characterised by the presence of mostly two or more metaxylem elements per fibrovascular bundle, has a similar anatomy to that of *Livistona* spp. palm tissue. Type B, represented largely by the peripheral section of the palm stem, is characterised by the presence of only one metaxylem element per fibrovascular bundle. Type B is only found in the Phase 2 assemblage and may represent the peripheral xylem of Type A or another taxon of the Arecaceae. The apex, or 'heart', and pith, or whole young stem, of several palms present in open woodland and monsoon vine forest environments in western Arnhem Land can be consumed (apex: *Carpentaria acuminata, Hydriastele ramsayi*; apex and pith: *L. benthamii, L. humilis* and *L. inermis*)[30,35]. While the apex of these palms may be eaten raw or lightly roasted, the pith requires roasting for an extended period (~12 h) prior to pounding. This process removes the most fibrous elements from the otherwise starchy pith, making the carbohydrates within the pith readily available for consumption. While there are no distinguishing apical features on the fragments of stem from the Phase 2 assemblage, their size makes it impossible to securely identify them as basal (pith) stem.

## Discussion

In summary, the plant macrofossil assemblage from Phase 2 provides evidence for the consumption of a range of plant foods (~10) by the EMH population occupying Madjedbebe from 65–53 kya. These plant foods were foraged from open forest and woodland, and, to a lesser extent, monsoon vine forest, and aquatic environments, all of which were likely in proximity to the site during this phase[13]. These plant foods provided a variety of dietary macronutrients, comprising carbohydrates (USOs, palm stems and fruits), fats and proteins (pandanus kernels and other tree nuts). Some of these plant foods, such as the fruits and easily-extracted nuts, represent readily available, high-ranked resources. However, other plant foods present in the assemblage required varying levels of processing prior to consumption. This included the cooking (and peeling) of USOs and palm stems, likely the further pounding of palm pith, and the laborious extraction of *Pandanus spiralis* kernels. Furthermore, while there were no edible seeds recovered in the plant macrofossil assemblage, residue and usewear evidence from the Phase 2 grinding stone

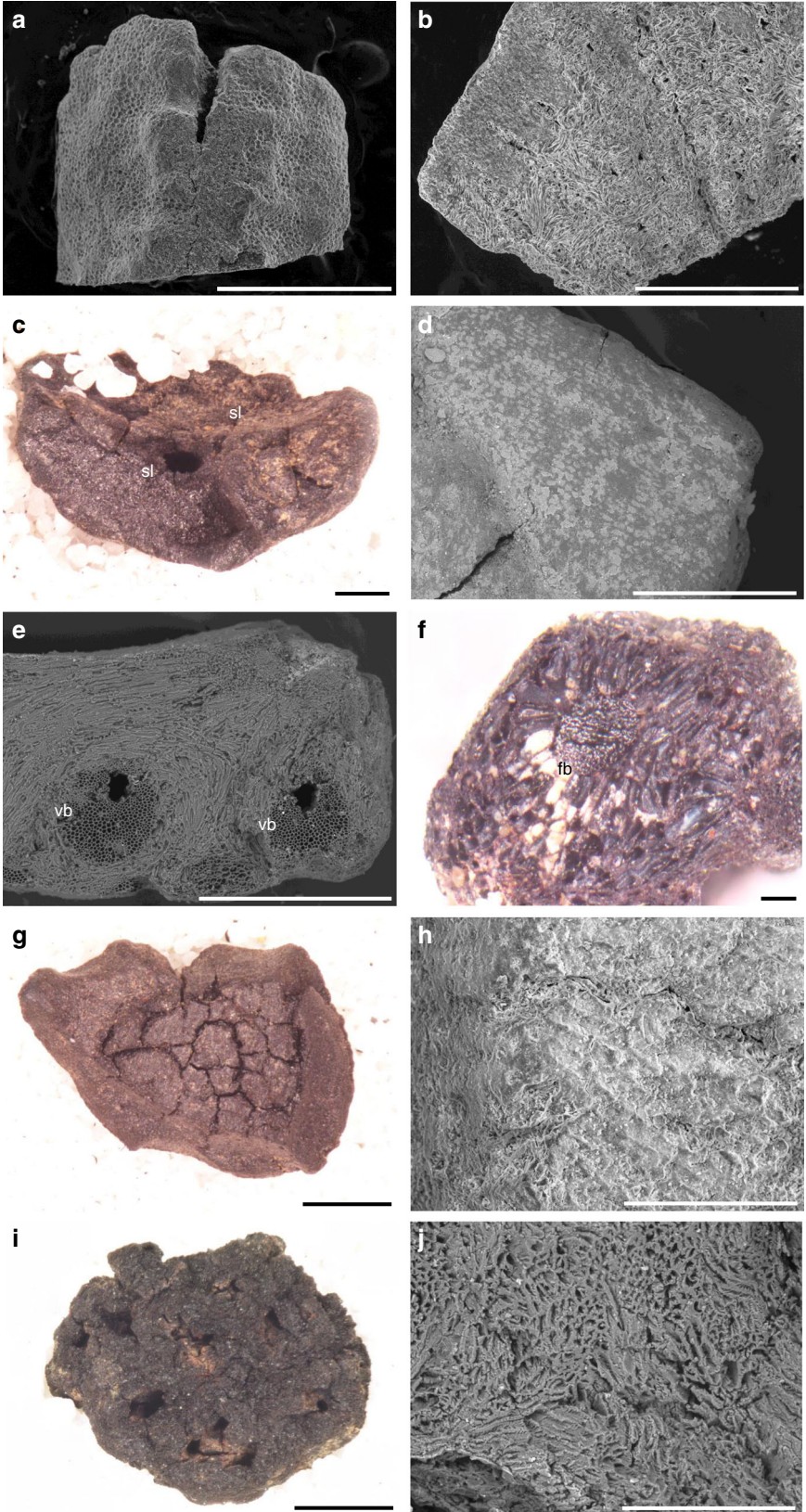

**Fig. 2 Examples of endocarp from Phase 2. a**, **b** *Buchanania* sp. endocarp from C2/46(HR), **a** scale bar is 1 mm, **b** transverse section, scale bar is 500 μm; **c** *Canarium australianum* endocarp from C2/38(HR), scale bar is 500 μm; **d** *C. australianum* endocarp from C2/37(HR), close-up of internal surface, scale bar is 500 μm; **e** Polydrupe *Pandanus* sp. endocarp from C2/42(HR), transverse section, scale bar is 500 μm; **f** *P. spiralis* mesocarp from C2/37, scale bar is 200 μm; **g**, **h** *Persoonia falcata* endocarp from C2/37(HR), **g** scale bar is 1 mm, **h** close-up of internal surface, scale bar is 200 μm; **i**, **j** *Terminalia* sp. endocarp from C2/37(HR), **i** scale bar is 1 mm, **j** transverse section, scale bar is 200 μm. See the supporting online information and Supplementary Figs. 3-7 for detailed identification proofs and the corresponding reference materials. sl seed locule, vb vascular bundle, fb fibrous bundle.

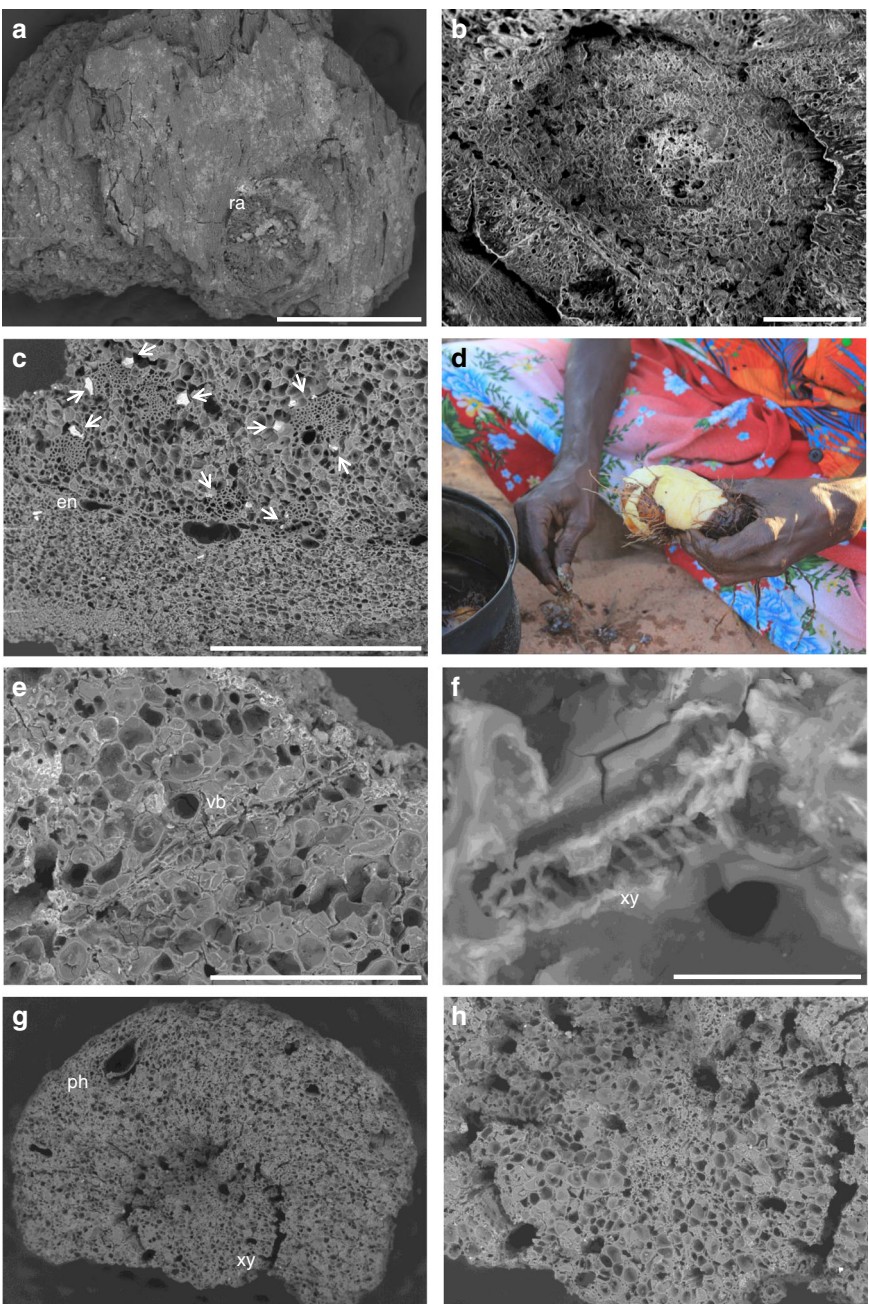

**Fig. 3 Examples of vegetative parenchyma from Phase 2. a**, **b** Monocotyledonous stem-based storage organ Type A from C2/41; **a** depicting skin-patterning and root abscission scar, scale bar is 500 μm; **b** close-up of root abscission scar, scale bar is 100 μm; **c** transverse section of monocotyledonous stem-based storage organ Type A from C2/32 A, depicting an endodermis and a series of closed collateral vascular bundles, arrows point to phytoliths on edges of vascular bundles, scale bar is 500 μm; **d** MN peeling a 'hairy' *Dioscorea bulbifera* tuber beside a hearth built to cook it, photo taken by SAF; **e**, **f** longitudinal section of monocotyledonous stem-based storage organ Type B; **e** scale bar is 300 μm; **f** close-up of vascular bundle, scale bar is 20 μm; **g**, **h** transverse section of secondary-root storage organ Type A from C2/39 A; **g** scale bar is 2 mm; **h** close-up of central tract of xylem, scale bar is 500 μm. ra root abscission scar, en endodermis, vb vascular bundle, ph phloem, xy xylem.

assemblage has identified seed-grinding during this initial phase of occupation[13]. While none of these plants are toxic, the processing required to extract and make edible the nutritious components from some of the taxa present is suggestive of multi-step and labour-intensive processing techniques.

These findings, which predate existing evidence for the processing of plant foods in Sahul by at least 23ky[36,37], suggest that a broader range of lower ranked plant foods was consumed during early occupation of Sahul than envisaged by proponents of the single coastal dispersal model[17,18]. This does not negate the possibility that EMH populations first entered Sahul exploiting high-ranked, coastal resources found on its now-submerged coastline. However, the investment of labour and technology into the extraction and processing of fruits, nuts, USOs and likely seeds at Madjedbebe 65–53 kya, does suggest that a broad diet was part of the toolkit employed by the EMH populations who reached Sahul.

Indeed, as EMH populations crossed the Wallace Line, novel fauna would likely have caused significant disruptions to their hunting strategies. However, many of the families, and in some

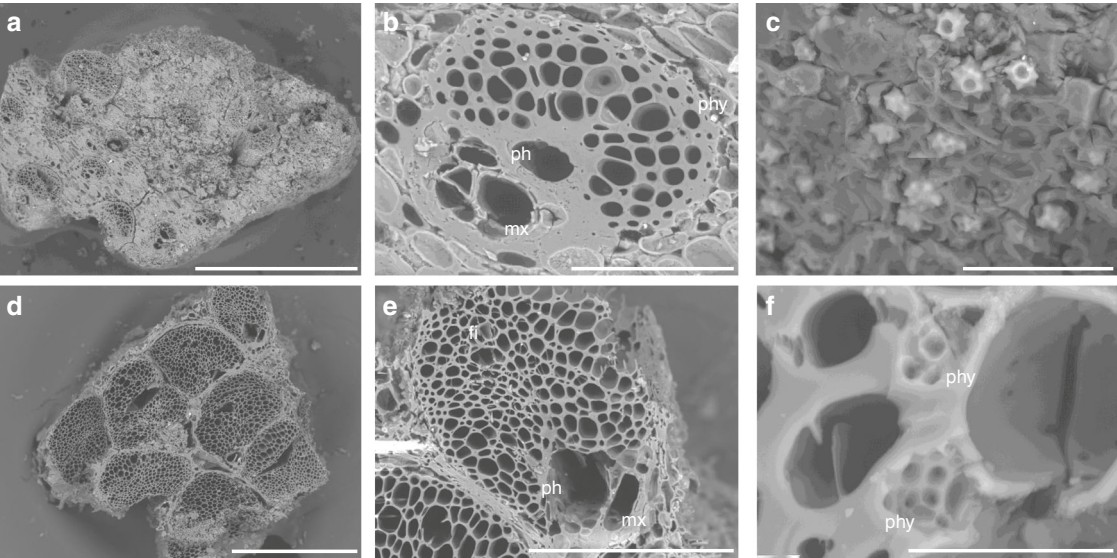

**Fig. 4 Examples of Arecaeae family stem tissue from Phase 2. a–c** Transverse section of Arecaceae stem Type A (cf. *Livistona* spp.) from C2/44, **a** scale bar is 1 mm, **b** close-up of fibrovascular bundle, scale bar is 100 µm, **c** close-up of globular echinate phytoliths, scale bar is 30 µm; **d, f** transverse section of Arecaceae stem Type B from C2/44, **d** scale bar is 500 µm, **e** close-up of fibrovascular bundle, scale bar is 200 µm; **f** transverse section of Arecaceae stem Type B from C2/45, close-up of globular echinate phytoliths, scale bar is 20 µm. See the supporting online information and Supplementary Fig. 8 for a detailed identification proof of Arecaceae stem Type A (cf. *Livistona* spp.) and the corresponding reference material. ph phloem, mx metaxylem, phy phytolith.

cases genera (including *Buchanania*, *Canarium*, *Livistona*, *Pandanus* and *Terminalia*), of plants available across the southern dispersal arc continue into Australia[38,39]. Therefore, the expenditure of labour in the preparation of a range of recognised plant foods could have ensured reliable access to fats, proteins and carbohydrates required to successfully move into the region.

The evidence for a broad plant food diet at Madjedbebe 65–53 kya is consistent with later Pleistocene archaeobotanical studies conducted in Island Southeast Asia and Sahul[8–12], and with evidence for EMH diets in Africa and the Middle East[4–6]. As such, it indicates that plant exploitation was a fundamental aspect of EMH diets globally. Culturally transmitted botanical knowledge, and the cognitive ability to perform multi-step and intensive processing sequences likely contributed to the adaptability and flexibility required by EMH populations to traverse continents and colonise new environments around the world.

## Methods

**Archaeobotanical analysis.** Two $1 \times 1$ m² columns (C3/1–27 and C2/28–57; C6/1–15 and C5/16–72) of excavated sediment and all hearths and other features identified during excavation were collected in their entirety for flotation (Supplementary Fig. 2c). A hundred percent of the sediment from these contexts underwent flotation, using a cascading 'Ankara-style' flotation tank[40].

All Phase 2 contexts in Square C2 (C2/46, C2/45, C2/44, C2/43, C2/42, C2/41, C2/40, C2/39 A, C2/38 A, C2/38, C2/37 A and C2/37) and a preserved hearth in Square C1 (C1/43 A) (Supplementary Fig. 2c) were analysed.

As flotation has been suggested to be less effective at recovering denser macrofossils, especially endocarp fragments[41], both the lighter fraction, or 'flot', and the heavy residue from these contexts were analysed. All charred fragments >1 mm from both fractions were analysed. The relevant plant macrofossils were sorted from the wood charcoal under low-powered light microscopy.

High-powered light microscopy and scanning electron microscope imaging was used to compare the anatomical and morphological features of the archaeological specimens to modern reference material from the region (see below). The identification of this archaeological material was limited by two things: the size of the modern reference collection; and the degree to which the botanical structures vary by family, genus and species.

No attempt was made to quantify the proportion of the diet contributed by different plants, analysis stopping simply at ubiquity. This is because many of the food plants that constituted the diet of people inhabiting the site at 65–53ka would not have been preserved archaeologically[42]. Indeed, many of the plant foods may have been eaten raw and have, therefore, been less likely to come into contact with

fire, or have been processed and/or eaten away from Madjedbebe as a part of a mobile foraging strategy[43]. This difference in preservation, determined by past human behaviour, would have been compounded by differences in the preservation rates and the level of identification possible for different taxa[44].

**Reference collection and ethnobotanical research.** The modern reference material was collected by SAF, MN, DjDj and research assistants over several seasons in Kakadu National Park. This was carried out with the permission of the Mirarr people, Gundjeihmi Aboriginal Corporation, Parks Australia and the Australian Government (Permit to carry out Scientific Research in a Commonwealth Reserve Permit No. RK870 and RK909; Access to Biological Resources in a Commonwealth Area for Non-Commercial Purposes Permit No. AU-COM2015-287, AU-COM2017-339, AU-COM2018-391). Alongside the production of a modern reference collection, ethnobotanical research was also undertaken with MN and DjDj, to define the material signature of plant exploitation practices in western Arnhem Land. Plants were identified in the field by MN and DjDj, and these identifications were then verified and furthered by the Northern Territory Herbarium.

Plant samples are housed in the University of Queensland Archaeobotanical Reference Collection, preserved dried, charred and in spirits. Following Hather 2000[34], where underground storage organs, stems and roots were part of the sample, stained thin-sections were produced (using a modified version of the method outlined by Johansen 1940[45]). This allowed for the anatomical structure of these plant parts to be understood prior to their transformation through charring.

**Reporting summary.** Further information on research design is available in the Nature Research Reporting Summary linked to this article.

## Data availability

All elements necessary to allow interpretation and replication of results, including full datasets and detailed archaeobotanical identification proofs are provided in the Supplementary Information. Data and R code for Supplementary Fig. 1 are online at https://doi.org/10.17605/OSF.IO/YDUZP[46]. Archaeobotanical material analysed in this study will be kept in the Archaeology Laboratories of the University of Queensland until 2021. It will then be deposited in a Gundjeihmi Aboriginal Corporation keeping place. The material will be publicly accessible upon request with permission from Gundjeihmi Aboriginal Corporation (gundjeihmi@mirarr.net) and the corresponding authors. The language, images and information contained in this publication includes reference to Indigenous knowledge including traditional knowledge, traditional cultural expression and references to biological resources (plants and animals) of the Mirarr people. The source Indigenous knowledge is considered "Confidential Information"; traditional law and custom applies to it and the Mirarr people assert ownership over it. Any Mirarr related language, images and information are published with the consent of Gundjeihmi Aboriginal Corporation as the representative of the Mirarr people for the purposes of education and specifically for use only in the context of this published work. Please

contact Gundjeihmi Aboriginal Corporation to request permission to refer to any Indigenous knowledge in this publication.

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

## Acknowledgements

The archaeobotanical and ethnobotanical research in this project was funded by a Wenner Gren Dissertation Fieldwork Grant (9260), an Australian Institute of Nuclear Science and Engineering Postgraduate Research Award (11877), a Dan David Scholarship and an Australian Postgraduate Award awarded to S.A.F. The initial fieldwork and excavation of Madjedbebe was funded by an Australian Research Council grant (DP110102864) obtained by C.C., B.M., R.F., M.S. and L.W. The authors are grateful to the custodians of Madjedbebe, the Mirarr Senior Traditional Owners (Yvonne Margarula and MN) and our research partner, the Gundjeihmi Aboriginal Corporation, for permission to carry out this research and publish this paper. We are also grateful to Justin O'Brien and David Vadiveloo for assistance in the field. We thank Dr Xavier Carah for implementing the archaeobotanical recovery program at Madjedbebe and for his assistance alongside Elspeth Hayes, Kasih Norman, Ashleigh Rogers, Makayla Harding and Kate Connell in the collection of plants and ethnobotanical data. This research was completed in part in Kakadu National Park under Permit No. RK870, RK909; however, the findings and views expressed are those of the authors and do not necessarily represent the views of Parks Australia, the Director of National Parks or the Australian Government. The authors acknowledge the assistance of the staff at Parks Australia, the Northern Territory Herbarium, and the University of Queensland's Archaeology, School of Earth Sciences, and Centre for Microscopy and Microanalysis Laboratories (especially Rachel Price, Linda Northdurft and Ian Cowie). We also thank the student volunteers from the University of Queensland for helping sort the heavy residue. Finally, the authors thank Quan Hua, Rachel Wood, Zenobia Jacobs, Elspeth Hayes and Alison Crowther for their advice and expert opinion during the drafting process.

## Author contributions

S.A.F. and A.F. conducted the archaeobotanical research. S.A.F., M.N. and Dj.Dj. conducted the ethnobotanical research and modern reference collection. S.A.F., A.F. and C.C. wrote the main text. M.N. and Dj.Dj. provided cultural knowledge and assisted in

the interpretation of the findings. S.A.F., B.M., C.C. and A.F. created the figures. C.C., R.F., B.M., M.S. and L.W. obtained the funding for and conducted the excavations.

## Competing interests

The authors declare no competing interests.
