## [Peer Review File · Nature Communications]

Reviewers' Comments:

Reviewer #1:

Remarks to the Author:

This paper presents the earliest archaeobotanical remains for the Sahul region, from the site of Madjedbebe in Arnhem Land, Northern Territory. It is rewarding to see this type of archaeobotanical analysis being applied to the assemblage from Madjedbebe. Too little is known of what Sahul's first colonists foraging strategies were, particularly in relation to plant remains. Their ability to learn about and adapt processing strategies for new plant foods is key to understanding the adaptability, innovation and flexibility of these first colonists, and this research is an excellent step towards provide this fuller picture of Sahul's earliest ancestors. At the same time, this research challenges the single dimensional coastal adaptation model previously suggested for the Southern Dispersal Route and colonization of Sahul.

At times the claims need to be better contextualized in further reading of the literature. In the abstract, the authors note that the use of plant foods requiring extensive processing did not occur in this region until the Holocene – yet there is evidence for toxic plant processing at Niah Cave in the late Pleistocene (see Barker et al 2007 *Journal of Human Evolution*) and then seed grinding from Australia at Lake Mungo from 25-14 kya (Fullagar et al 2015 *Archaeology in Oceania*) and at Cuddie Springs from 30 kya (Fullagar and Field 1997 *Antiquity*). The claim that the practices at Madjedbebe of being more than 40kya older than other such finds in Australia is perhaps a little exaggerated, and also does not take into consideration the age range of the Phase 2 finds so changing to 23-35 kya may be more accurate.

Further consideration also needs to be considered as to the language used so that conclusions are not over-stated: are the plant resources identified in this article really "hard to get" or are they just time consuming in processing? They do not appear to be toxic – or are they restricted in range/abundance/difficult to obtain? I would apply the same critique to the use of the term "broad-spectrum". Are five taxa indicative of broad spectrum plant exploitation? Compared to the more numerous species at Niah especially. The Ivane valley is also used to support broad-spectrum plant exploitation but there are only two plant species *Pandanus* and *Dioscorea* published for this site? Are there ethnographic correlates that could be drawn upon here? For example, how does this diet compare to the ethnobotanical information collected as part of this project?

Overall the data for the identification of the archaeobotanical remains is presented well and is convincing, particularly the proofs – although there are some errors. Page 2 Line 66 – Eight different types of plants were recovered from Phase 2 – unclear what is meant by different types here – taxa? Or types of plant remains? Page 2 Line 66 – should this be 5 taxa? *Curcuma*, *Buchanania*, *Canarium*, *Pandanus* and *Persoonia*? Page 4, line 100 – Three taxa should be Four.

Page 4, paragraph 3 and also supplementary Information Section 3 – consistency with plant terminology needed as there is switching between drupes, nuts and kernels, particularly for the experiment which makes it unclear as to the specific activity being undertaken. Perhaps the overall fruit should be identified as a syncarp, then polydrupe or drupe, and kernels for the actual nuts itself? See Lim 2012 *Edible Medicinal and Non-Medicinal Plants Volume 4: Fruits*.

Further information is also required to support some of the statements made. For example "reliable access to fats, proteins and starches in a range of habitats". There is no table included anywhere that identifies the dietary/nutritional (or other medicinal) properties of these taxa, nor is there any discussion as to what types of habitats that the plants come from. *Cucurma* in particular also has medicinal properties (see Rajkumari and Sanatombi 2018 *International Journal of Food Properties*) –

was this apparent in the ethnobotanical knowledge recorded? Or for any of the other species identified?

Further to the point above, there is no palaeoenvironmental evidence included that discusses the availability of the same vegetation types today as in the past. Or how these vegetation zones have changed over time. Is this a potential factor in identifying archaeobotanical remains from the site?

The authors have also noted that Phase 2 is the earliest evidence for human occupation at the site, whereas in previous articles (Clarkson et al 2017), there appears to be lithic artefacts at least in Phase 1. Is it clear that there is no evidence for occupation in Phase 1? Are there any plant remains that belong to this phase?

In addition to the archaeobotanical remains, usewear/residue analysis of 29 grinding stones is included to assess plant processing at the site. These are very significant artefacts – the oldest in Sahul and identifying their use provides another important aspect to our understanding of the adaptations and behaviours of Sahul's earliest colonists. However, this section needs to be presented more coherently. To begin with, there are inconsistencies in the total grinding stone numbers used for plant processing – in the text of the article, there is at least 8. In SI Section 4, it ranges between 10 (page 1, Line 129), then 10 in the table if you include the two possibles, then page 2 line 167, there are 7, and then in SI Table 6, Line 266 there are 15. In SI Section 4, page 3, line 197 – only 5 artefacts had plant residues attributed to use. While I appreciate that there are multiple strands of evidence being used to analyse these tools, some clarity or resolution is needed for the reader where the results of these different analyses are contrasted to either support or reject the tool function.

In the Clarkson et al 2017 article GS32 is identified as a mortar used to process hard plant material. Yet this is not discussed in the body of this paper. It is included in the SI Section 4, although if it is used as a mortar there appears to be no impact marks in the usewear (Supp Table 4).

There needs to be photographic proofs of experimental/replicative tools used for the suggested use activities, particularly for the seed grinding. This level of information is included for the archaeobotanical identifications but not for the usewear/residues. What range of experiments/activities were undertaken as a comparison for the archaeological tools? This is an important section missing from both this and earlier Madjedbebe reports attributing function to specific tools. For example, *P. spiralis* has been recorded ethnographically as ground into a flour – is this a potential use that was tested for? At this point, the reader has to take on face value the identification of these tools as there is no conclusive proofs given as to their function.

Another worrying issue is that the authors state that pigments appears to be have resulted on the grinding stones due to post-depositional or post-excavation handling. Is there a possibility that this has occurred with other types of residues, particularly as it appears the animal residues are discounted? Again, this comes back to the need to integrate all of the strands of evidence into a more coherent picture.

For what grinding stones is there deliberate evidence of shaping? More information is required here.

Overall this is an extremely important site in Sahul's prehistory and the identification of the activities undertaken by our earliest colonists is vital for understanding colonizing strategies, not just for Sahul, but for the global domination by modern humans. The focus on plant remains broadens our view of these colonists, challenging single dimensional models such as pre-adaptations to Coastal Highways. At the same time, this paper shows the need for these types of multidisciplinary approaches to site interpretation and is hoped will influence similar types of studies on other sites. The data here is

important but needs to be presented in more coherent and conclusive detail to fully justify the conclusions made.

These comments are largely informal and to do with spelling/grammar issues:

1. Page 7 Figure 4, line 178 c-d should just be c
2. Page 9, line 285 – spelling of remains
3. Extended Data Figure 1 – 1972 excavation in figure, 1973 excavation in Caption.
4. Extended Data Figure 3b – is it possible for a clearer picture?
5. SI Section 1 line 48 – space between Shelf and (Nix
6. SI Section 1 line 63 – spelling Zingiberaceae
7. SI Section 2 – include total NISP
8. SI Section 4, page 2, line 155 – stone will cause
9. SI Section 4, page 2, line 159 concave usually indicates
10. SI Section 4, page 3, line 172 – micro-striations
11. SI Section 4, page 3, p187 - ultrasonicated
12. Page 2 Line 72 – spelling of specifically
13. SI Section 4, page 3, line 173 - the four pigment processing stones are not discussed here and appear to be included in the 21 non-diagnostic artefacts.

Reviewer #2:

Remarks to the Author:

The manuscript by Florin et al. presents evidence of plants exploited by the first human inhabitants of Sahul, purportedly 65-53kya. The authors further note that some of the plant resources exploited at the Madjedbebe rockshelter required labour-intensive processing, thus predating previous evidence for this behaviour in Australia by >40ky. This study has potential to influence archaeological narratives in Australia and beyond and merits publication in Nature Communications, but I feel that some of the claims made by the authors are not sufficiently justified or need further clarification. For this reason, I am recommending a major review. Below I describe the points that, in my opinion, the authors would need to address in order to improve the manuscript.

- The manuscript has no introduction whatsoever; it goes straight from the abstract to the presentation of the results. Although Nature Communication's guide for authors advises against the use of 'Introduction' as a heading, this should not preclude the authors from discussing previous literature and contextualising their research.
- It is not clear what the actual focus of the manuscript is. If it is the early evidence for the consumption of labour-intensive plant resources this should be clearly stated at the beginning of the manuscript and the relevant literature (similar evidence for early modern humans worldwide) discussed. The groundstone assemblage and the experimental work would also need to be further discussed in the main text, as they provide supporting data for the central claim.
- A more robust chronological framework seems necessary to support the authors' claims. In the first place, the disagreement surrounding the date of Madjedbebe's Phase 2, most notably by Allen (2017) and O'Connell et al. (2018), should at least be briefly acknowledged. Clarkson et al. (2018), in a response to Allen's (and others) (2017) comments, show that beyond 1.5m of depth 75% (9 out of 12) of the charcoal samples submitted for radiocarbon dating did not survive chemical pretreatment (Clarkson et al. 2018: Fig. 1), but in this manuscript Florin et al. claim that "attempts to date the plant macrofossil remains from Phase 2 have been unsuccessful, dissolving during chemical pretreatment"

(p. 6, l. 6-7). Whether all charcoal samples from Phase 2 were destroyed during pretreatment or three did survive needs, thus, further clarification here. Indeed, 14C-dated charcoal fragments from Phase 2 presented in Clarkson et al. (2017: Extended Data Figure 8(g)) seem to disagree with the 65-53kya OSL ages. Did any of the 'disagreeing' 14C dates come from the Phase 2 hearth? Also, have there been any attempts to radiocarbon date macrobotanical remains other than charcoal? A ≥ 50 kya 14C date on two or more of the plant food remains presented in this study would most definitely provide definite evidence to the consumption of labour-intensive resources by the first human inhabitants of Sahul.

- Despite its potential, the paragraph describing the groundstone assemblage in the main text, including use-wear and residue analyses, adds little to the overall discussion. It is not clear how the presented evidence is related to the macrobotanical assemblage. The authors merely state that at least 8 out of 29 tools were used for plant processing, including seed-grinding (n=2) and pounding (n=1). If the authors wish to establish a direct correlation between the use of the grinding stones and the consumption of labour-intensive plant resources, as seems to be the case, more clear evidence needs to be presented. In particular, starch grain analysis has the potential to provide direct evidence of the processing of USOs, nuts and seeds. According to SI Section 4, high densities of starch grains were encountered in at least two grinding stones. Have these starch grains been identified? Were *Buchanania* sp., *Canarium australianum*, *Curcuma australasica*, *Pandanus spiralis* and *Persoonia falcata* included in the modern starch reference collection? Have starch grains from these taxa been identified in the grinding stones from Madjedbebe? If this research is still ongoing and/or the authors do not wish to present it here they may reconsider establishing a direct correlation between the groundstone assemblage and the consumption of labour-intensive plant resources until more conclusive evidence is available.

Juan José García-Granero

References

- Allen, J. 2017. Yes, Virginia, there is a Santa Claus; He just doesn't bring presents to children who don't believe in him. *Australian Archaeology* 83:163–165.
- Clarkson, C., et al. 2017. Human occupation of northern Australia by 65,000 years ago. *Nature* 547:306–310.
- Clarkson, C. et al. 2018. Reply to comments on Clarkson et al. (2017) 'Human occupation of northern Australia by 65,000 years ago'. *Australian Archaeology* 84:84–89.
- O'Connell, J.F. et al. 2018. When did *Homo sapiens* first reach Southeast Asia and Sahul?. *Proceedings of the National Academy of Sciences* 115:8482–8490.

Below are our responses to relevant reviewers' comments. Comments have been responded to in three ways:

- 1) The majority have been accepted;
- 2) In response to Reviewer #1 and #2's concerns with the residue and usewear analysis presented, we have decided to follow Reviewer #2's advice and omit this secondary source of data. Therefore, comments on the residue and usewear analysis have not been responded to, as they are no longer applicable to the current paper; and
- 3) We have rejected Reviewer #2's assertion that a "more robust chronological framework" is necessary in order to publish these results. Madjedbebe's initial phase of dense occupation (Phase 2, c.65-53kya) is beyond the limit of AMS radiocarbon dating and has, therefore, been dated with Optically-Stimulated Luminescence (OSL). The results of this extensive and double-blinded dating regime have been published by Clarkson et al. (*Nature* 2017).

Reviewer #1 (Remarks to the Author):

- In the abstract, the authors note that the use of plant foods requiring extensive processing did not occur in this region until the Holocene – yet there is evidence for toxic plant processing at Niah Cave in the late Pleistocene (see Barker et al 2007 *Journal of Human Evolution*) and then seed grinding from Australia at Lake Mungo from 25-14 kya (Fullagar et al 2015 *Archaeology in Oceania*) and at Cuddie Springs from 30 kya (Fullagar and Field 1997 *Antiquity*). The claim that the practices at Madjedbebe of being more than 40kya older than other such finds in Australia is perhaps a little exaggerated, and also does not take into consideration the age range of the Phase 2 finds so changing to 23-35 kya may be more accurate.

Accepted. We agree with this assessment of the literature and have changed this statement to, "Our finds predate existing evidence for such subsistence practices in Sahul by at least 23ky." (Page 1, Lines 36-37; see also Page 9, Lines 7-8)

- Further consideration also needs to be considered as to the language used so that conclusions are not over-stated: are the plant resources identified in this article really "hard to get" or are they just time consuming in processing?

Accepted. We have amended this language and have replaced "the expenditure of labour on hard to get plant foods" with "the expenditure of labour on the processing of plants". (Page 1, Line 39)

- I would apply the same critique to the use of the term "broad-spectrum". Are five taxa indicative of broad spectrum plant exploitation? Compared to the more numerous species at Niah especially. The Ivane valley is also used to support broad-spectrum plant exploitation but there are only two plant species *Pandanus* and *Dioscorea* published for this site?

Accepted. We agree that the term "broad-spectrum" may not be the best language to be used for this study, as it usually suggests a diachronic

increase in exploited resources. We have amended the introduction and conclusions to reflect this and the term broad-spectrum is no longer used.

We have discussed the evidence for EMH plant diets in the introduction, further clarifying the importance of archaeobotanical analysis at Niah Cave and Kosipe.

“Recent research into plant macro- and micro-fossils is beginning to break down this paradigm: the use of plant foods, including those associated with later agricultural transitions, such as grass seeds and underground storage organs (USOs), is now evidenced in Middle Stone Age sites in Africa and the Middle East²⁻⁵; the processing of toxic plants (*Dioscorea hispida* and *Pangium edule*) is now dated to as early as 46kya in Niah Cave, Borneo⁶⁻⁸; the translocation of yams (*Dioscorea* spp.) to high altitudes and management of monodrupe pandanus stands, facilitated early use of highland environments in New Guinea (~49kya)^{9,10}; and associated plant-processing technologies, such as nut-crackers and seed-grinding stones, are linked to EMH dispersals into the Middle East and Australia^{11,12}.” (Page 2, Lines 4-13)

And, we have rewritten the results section to reflect improvements in the archaeobotanical analysis since the previous submission. “Genus-, or species-specific identifications have been made for five taxa, with a further five types identified to broader botanical categories.” (Page 3, Lines 13-16)

- Page 2 Line 66 – Eight different types of plants were recovered from Phase 2 – unclear what is meant by different types here – taxa? Or types of plant remains?

Accepted. The concept of ‘types’ has been clarified. “Genus-, or species-specific identifications have been made for five taxa, with a further five types identified to broader botanical categories.” (Page 3, Lines 13-16)

- Page 2 Line 66 – should this be 5 taxa?

Accepted. This has been changed to five.

- Page 4, line 100 – Three taxa should be Four.

Accepted. This has been changed to, “Charred endocarps of five fruit and nut taxa were identified: *Buchanania* sp.; *Canarium australianum*; polydrupe *Pandanus* sp.; *Persoonia falcata*; and *Terminalia* sp. (Fig. 2).” (Page 4, Lines 2-4)

Further archaeobotanical analysis allowed for the identification of another taxon (*Terminalia* sp.).

- Page 4, paragraph 3 and also supplementary Information Section 3 – consistency with plant terminology needed as there is switching between

drupes, nuts and kernels, particularly for the experiment which makes it unclear as to the specific activity being undertaken. Perhaps the overall fruit should be identified as a syncarp, then polydrupe or drupe, and kernels for the actual nuts itself? See Lim 2012 Edible Medicinal and Non-Medicinal Plants Volume 4: Fruits.

Accepted. We agree and have changed our terminology to reflect Lim 2012.

- Further information is also required to support some of the statements made. For example “reliable access to fats, proteins and starches in a range of habitats”. There is no table included anywhere that identifies the dietary/nutritional (or other medicinal) properties of these taxa.

Accepted. We have included discussion of the nutritional properties of the plant foods. However, as in many cases there is no data for the Australian species in question or our botanical types are too broad for the data to be of use, we have largely kept this discussion broad:

- *Pandanus spiralis*: “The small kernels are rich in fat (44-50%) and protein (20-34%)³⁶.” (Page 4, Lines 28-29)
- All: “These plant foods provided a variety of dietary macronutrients, comprising carbohydrates (USOs, palm stems and fruits), fats and proteins (pandanus kernels and other tree nuts).” (Page 8, Lines 29-31)

None of the plants identified are primarily medicinal in nature (*Curcuma australasica* was misidentified in the earlier submission). As such we have, in discussion with the Gundjeihmi Aboriginal Corporation and Mirarr elders involved, decided to not publish any additional medicinal information about these plants. This has been done to protect the Mirarr peoples’ intellectual property.

- Nor is there any discussion as to what types of habitats that the plants come from.

Accepted. We have included discussion of the habitats of all identified taxa and one other ‘type’:

- *Buchanania* sp.; *Canarium australianum*; polydrupe *Pandanus* sp.; *Persoonia falcata*; and *Terminalia* sp.: “All of these taxa are common in open forest and woodland, and/or monsoon vine forest environments³².” (Page 4, Lines 4-5)
- Polydrupe *Pandanus* sp.: “There are two types of polydrupe pandanus in Arnhem Land: *P. spiralis* and *P. basedowii*³². Only one fragment from Madjedbebe, a portion of mesocarp, can be securely identified as *P. spiralis* (Fig. 3f) but it is likely that the majority come from this variety since *P. basedowii* only grow on the escarpment top and would have been difficult to access from Madjedbebe (see Fig. 1b).” (Page 4, Lines 18-22).
- Monocotyledonous stem-based storage organ Type A: “The presence of an endodermis, in a larger fragment of this type recovered from

Phase 3, allows for its further identification as an aquatic or semi-aquatic USO (Fig. 3c). Endodermis is rarely present in stem tissue, except in aquatic plants where they are significant in controlling water balance within perennating stems³⁷.” (Page 6, Lines 14-18)

- Further to the point above, there is no palaeoenvironmental evidence included that discusses the availability of the same vegetation types today as in the past. Or how these vegetation zones have changed over time. Is this a potential factor in identifying archaeobotanical remains from the site?

Accepted. We have now added a sentence in the discussion addressing this: “These plant foods were foraged from open forest and woodland, and, to a lesser extent, monsoon vine forest, and aquatic environments, all of which were likely in proximity to the site during this broadly cooler and wetter phase¹².” (Page 8, Lines 26-29)

- The authors have also noted that Phase 2 is the earliest evidence for human occupation at the site, whereas in previous articles (Clarkson et al 2017), there appears to be lithic artefacts at least in Phase 1. Is it clear that there is no evidence for occupation in Phase 1? Are there any plant remains that belong to this phase?

Accepted. As described in Extended Data Fig.1a, there is a sharp drop-off in charred plant macrofossil recovery in Phase 1. Whilst flotation recovered 1048 plant macrofossils from ~507 Litres of soil in Phase 2, it only recovered 49 plant macrofossils from ~401 Litres of soil in Phase 1. Further, ~50% of the Phase 1 plant macrofossils come from the two spits directly below Phase 2 and are comprised of the same taxa. In keeping with Clarkson et al. 2017 we have changed the wording to state that Phase 2 is “the earliest layer of dense occupation at Madjedbebe.” (Page 2, Lines 32-33)

- There are inconsistencies in the total grinding stone numbers used for plant processing – in the text of the article, there is at least 8. In SI Section 4, it ranges between 10 (page 1, Line 129), then 10 in the table if you include the two possibles, then page 2 line 167, there are 7, and then in SI Table 6, Line 266 there are 15. In SI Section 4, page 3, line 197 – only 5 artefacts had plant residues attributed to use.

No longer applicable.

- In the Clarkson et al 2017 article GS32 is identified as a mortar used to process hard plant material. Yet this is not discussed in the body of this paper. It is included in the SI Section 4, although if it is used as a mortar there appears to be no impact marks in the usewear (Supp Table 4).

No longer applicable.

- There needs to be photographic proofs of experimental/replicative tools used for the suggested use activities, particularly for the seed grinding. This level of information is included for the archaeobotanical identifications but not for the usewear/residues. What range of experiments/activities were undertaken as a comparison for the archaeological tools?

No longer applicable.

- Another worrying issue is that the authors state that pigments appears to be have resulted on the grinding stones due to post-depositional or post-excavation handling. Is there a possibility that this has occurred with other types of residues, particularly as it appears the animal residues are discounted?

No longer applicable.

- For what grinding stones is there deliberate evidence of shaping?

No longer applicable.

- Page 7 Figure 4, line 178 c-d should just be c

No longer applicable.

- Page 9, line 285 – spelling of remains

Accepted. This has been corrected.

- Extended Data Figure 1 – 1972 excavation in figure, 1973 excavation in Caption.

Accepted. This has been corrected.

- Extended Data Figure 3b – is it possible for a clearer picture?

Accepted. This picture has been replaced.

- SI Section 1 line 48 – space between Shelf and (Nix

Accepted. This has been corrected.

- SI Section 1 line 63 – spelling Zingiberaceae

Accepted. Zingiberaceae is no longer written in this section.

- SI Section 2 – include total NISP

Accepted. Total NISP has been added.

- SI Section 4, page 2, line 155 – stone will cause
No longer applicable.
- SI Section 4, page 2, line 159 concave usually indicates
No longer applicable.
- SI Section 4, page 3, line 172 – micro-striations
No longer applicable.
- SI Section 4, page 3, p187 – ultrasonicated
No longer applicable.
- Page 2 Line 72 – spelling of specifically
Accepted. This has been corrected.
- SI Section 4, page 3, line 173 - the four pigment processing stones are not discussed here and appear to be included in the 21 non-diagnostic artefacts.
No longer applicable.

Reviewer #2 (Remarks to the Author)

- The manuscript has no introduction whatsoever; it goes straight from the abstract to the presentation of the results. Although Nature Communication's guide for authors advises against the use of 'Introduction' as a heading, this should not preclude the authors from discussing previous literature and contextualising their research.

Accepted. An introduction has now been added to the paper. We apologise to the reviewer for not having included one originally. This was because the paper had been written in the format of a Letter to Nature not a Nature Communications article.
- It is not clear what the actual focus of the manuscript is. If it is the early evidence for the consumption of labour-intensive plant resources this should be clearly stated at the beginning of the manuscript and the relevant literature (similar evidence for early modern humans worldwide) discussed. The groundstone assemblage and the experimental work would also need to be further discussed in the main text, as they provide supporting data for the central claim.

Accepted. We have refocused the paper with a new introduction looking at the role of plants in early modern human diets and the importance of diet for understandings of the southern dispersal out of Africa.

- A more robust chronological framework seems necessary to support the authors' claims. In the first place, the disagreement surrounding the date of Madjedbebe's Phase 2, most notably by Allen (2017) and O'Connell et al. (2018), should at least be briefly acknowledged. Clarkson et al. (2018), in a response to Allen's (and others) (2017) comments, show that beyond 1.5m of depth 75% (9 out of 12) of the charcoal samples submitted for radiocarbon dating did not survive chemical pretreatment (Clarkson et al. 2018: Fig. 1), but in this manuscript Florin et al. claim that "attempts to date the plant macrofossil remains from Phase 2 have been unsuccessful, dissolving during chemical pretreatment" (p. 6, l. 6-7). Whether all charcoal samples from Phase 2 were destroyed during pretreatment or three did survive needs, thus, further clarification here. Indeed, 14C-dated charcoal fragments from Phase 2 presented in Clarkson et al. (2017: Extended Data Figure 8(g)) seem to disagree with the 65-53kya OSL ages. Did any of the 'disagreeing' 14C dates come from the Phase 2 hearth? Also, have there been any attempts to radiocarbon date macrobotanical remains other than charcoal?

Rejected. Radiocarbon dating is not an appropriate method by which to date Phase 2 at Madjedbebe. First, the age range of 65,000–53,000 years is beyond that of the AMS radiocarbon time scale. Second, as acknowledged, the charred plant macrofossil samples have proven to be chemically altered with depth (Clarkson et al. *Australian Archaeology* 2018) and are not likely to yield accurate results.

The three radiocarbon dates described in Clarkson et al. (*Australian Archaeology* 2018) to have survived chemical pre-treatment are from Phase 3, not Phase 2, and are published in Clarkson et al. *Nature* 2017. All charcoal submitted for dating from the hearth in Phase 2 dissolved during the base stage of ABOX pre-treatment.

The increased chemical, but not structural, alteration of charcoal with depth evidenced at the site is consistent with the antiquity of the early plant macrofossil assemblage; charcoal likely transformed to humic substances over time via a process of oxidative degradation (Ascough et al. *Journal of Archaeological Science* 2011). This is not an isolated phenomenon. The pervasive chemical alteration of charcoal, which retains structural features, is prevalent in tropical sites (Wood *Australian Archaeology* 2017) and was also reported at the nearby archaeological site of Nauwalabila I (Fifield et al. *Radiocarbon* 2016; Bird et al. *Quaternary Science Reviews* 2002). This phenomenon is already published in Clarkson et al. (*Australian Archaeology* 2018) and is not the topic of this paper.

As radiocarbon dating is not an appropriate method by which to date Phase 2 at Madjedbebe we have relied on the OSL chronology already published for

this site in Clarkson et al. (Nature 2017). This chronology is based on an extensive (56 samples) and double-blinded, single-grain OSL dating regime. It has clear internal consistency, both within the OSL samples, and between the OSL samples and the AMS radiocarbon samples, where the latter are technically possible in a tropical site (up to ~34,000 years cal BP). This dating regime has been reviewed and supported by several leading AMS radiocarbon-dating specialists, including Quan Hua at the Australian Nuclear Science and Technology Organisation (Clarkson et al. *Nature* 2017; Clarkson et al. *Australian Archaeology* 2018) and Rachel Wood at the Australian National University (Wood *Australian Archaeology* 2017).

There is an increasing body of genetic and fossil evidence to suggest the global spread of anatomically modern humans occurred pre-65kya (e.g. Bae et al. *Science* 2017; Demeter et al. *PLoS One* 2015; Liu et al. *Nature* 2015; Prüfer et al. *Science* 2017; Westaway et al. *Nature* 2017) and no evidence to support the assertion presented in O'Connell et al. (*PNAS* 2018), that the over 10,000 artefacts represented by Phase 2 at Madjedbebe moved downwards by ~45cm-60cm. Instead, the change in both lithic industry and raw material between Phase 2 and Phase 3 at Madjedbebe, and the limited dispersion of equivalent dose values in the single-grained OSL samples, provide evidence that the Pleistocene layers of the site saw minimal disturbance (Clarkson et al. *Nature* 2017).

- Despite its potential, the paragraph describing the groundstone assemblage in the main text, including use-wear and residue analyses, adds little to the overall discussion. It is not clear how the presented evidence is related to the macrobotanical assemblage. The authors merely state that at least 8 out of 29 tools were used for plant processing, including seed-grinding (n=2) and pounding (n=1). If the authors wish to establish a direct correlation between the use of the grinding stones and the consumption of labour-intensive plant resources, as seems to be the case, more clear evidence needs to be presented. In particular, starch grain analysis has the potential to provide direct evidence of the processing of USOs, nuts and seeds. According to SI Section 4, high densities of starch grains were encountered in at least two grinding stones. Have these starch grains been identified? Were *Buchanania* sp., *Canarium australianum*, *Curcuma australasica*, *Pandanus spiralis* and *Persoonia falcata* included in the modern starch reference collection? Have starch grains from these taxa been identified in the grinding stones from Madjedbebe? If this research is still ongoing and/or the authors do not wish to present it here they may reconsider establishing a direct correlation between the groundstone assemblage and the consumption of labour-intensive plant resources until more conclusive evidence is available.

No longer applicable. The residue and usewear analysis is secondary to the archaeobotanical analysis of the plant macrofossil assemblage and is not ready for publication. As such, we agree with your conclusion and have omitted this data. The absence of this evidence does not affect the key claims made within this paper or diminish the significance of this unique assemblage,

which provides the earliest evidence for modern human diet in Sahul and one of the few opportunities to explore the role of plants in early modern human diet and dispersal globally.

Reviewers' Comments:

Reviewer #1:

Remarks to the Author:

The paper has addressed many of the concerns raised by the initial review. As such it reads as a stronger paper with particular focus upon the archaeobotanical remains and is an important contribution to our understanding of diet, resource use and colonizing strategies of modern humans. However, there are some issues with some of the changes made that need to be addressed.

The article has brought in additional evidence of other studies completed on EMH plant use to provide a context for their own study. I am concerned with two of these that relate to Geshar Benot Ya'aqov in Israel (references 4 and 12) – this is an Acheulian site that dates to 780,000 years ago and does not relate to EMH and should be removed.

Ten plant types are referred to on page 3 line 16 and again on page 8 line 25. This would be *Buchanania*, *Canarium australianum*, polydrupe *Pandanus* sp.; *Persoonia falcata* and *Terminalia* sp (from endocarps); Monocotyledonous stem-based storage organ types A and B (from vegetative parenchyma); two types of *Areceaceae* (stem tissue) and the tenth is seeds based on grinding stone evidence? This needs to be made clearer particularly as the article has now removed the grinding stone evidence and there appears to be no seeds found within the archaeobotanical remains. Perhaps discussing why there is a lack of seeds also needs to be considered particularly due to the grinding stone evidence.

One of the examples identified as *Curcuma australasica* rhizome in the first submission is now identified as *Areceaceae* stem tissue.

Reviewer #3:

Remarks to the Author:

The manuscript presents the findings of some excellent archaeobotanical research at the EHM site of Madjedbebe in NW Australia. Although the age of the Phase 2 occupation presented here is questioned by some, it can be presumed to minimally predate 50K bp. As such, the materials under discussion here - especially those derived from the fills of archaeological features - are mostly associated with EHM occupation of the site.

The archaeobotanical research is very impressive and conservative in terms of the identifications, which is always welcome. I found the SOM for identifications to be excellent and convincing in terms of rationale for taxonomic identifications.

I have four minor issues that the authors may wish to consider:

1 - The argument at the beginning that EMH dispersal is more associated with faunal exploitation than floral exploitation does not really ring true. Despite the paucity of sites, and the lack of robust archaeobotanical recovery at most, some of the oldest EMH sites in Sunda and Sahul are clearly associated with plant exploitation, including of toxic foods. Maybe start the argument with a different emphasis - these new findings at Madjedbebe reinforce the findings from Niah and Ivane Valley - and present robust evidence for the exploitation of a more diverse plant assemblage than at any other early site. I would take this to be the key point.

Similarly, suggested broadening of diet in Holocene and association with agriculture is a very old idea

- the reference used is Flannery 1969 - actually agriculture likely leads to restriction of diet in many places. So, I think this is a false start to get the argument going - it would be much better to start with early evidence from MSA sites in Africa, Niah and Ivane - and go from there ...

2 - Although the plants identified require processing and cooking - these are not as necessarily labour intensive as suggested in the manuscript. The archaeobotany of EMH occupation at Niah suggests toxic plant food processing, whereas this is not necessarily implied by the Madjedbebe data. So, perhaps tone this aspect down.

3 - Pandanus extraction experiment is completely out-of-place and nonsensical - no doubt based on urban dwellers who have no experience with extracting materials, other than cracking the odd walnut or peeling a banana. There is no comparison - as a skilled practitioner could probably do work with Pandanus as easily as we would do these mundane activities with walnuts and bananas. I would suggest omitting the '54 blows' sentence and associated SOM. Just think - how long would it take most people to milk a cow relative to someone who knows what they are doing?

4 - Antiquity of plant remains is a thorny issue. As I understand it, most of these materials cannot be directly radiocarbon dated because they are at/beyond the range of the technique. Ok. However, that leaves the issue of archaeological association and antiquity unresolved for those archaeobotanical fragments distributed through the stratigraphy, rather than those collected from the hearth (which are presumably more secure in terms of chronostratigraphic association). Surely a more relevant SOM than the Pandanus extraction experiment would be to look at the age-depth comparison for the site as a whole between radiocarbon and OSL dates - showing how the ability to date carbon-based materials (such as older plant fragments) tails off downward. I also wonder if there is carbon replacement in some archaeobotanical remains - especially more porous parenchyma - which could actually be measured directly in those fragments to show that there is increasing replacement of carbon with depth; hence, meaning they cannot be directly dated (even if not beyond the radiocarbon limit). Just a thought.

Lastly - use of 'bush foods' in title is somewhat misleading - as only has currency in Australian context - even if it will play well in the media.

We thank the editor and reviewers for their critique. The following paragraphs respond to the critique seriatim. Changes and additions are shown in both a tracked version and a clean version of the revised text.

Reviewer #1 (Remarks to the Author):

The paper has addressed many of the concerns raised by the initial review. As such it reads as a stronger paper with particular focus upon the archaeobotanical remains and is an important contribution to our understanding of diet, resource use and colonizing strategies of modern humans. However, there are some issues with some of the changes made that need to be addressed:

1. The article has brought in additional evidence of other studies completed on EMH plant use to provide a context for their own study. I am concerned with two of these that relate to Gesher Benot Ya'aqov in Israel (references 4 and 12) – this is an Acheulian site that dates to 780,000 years ago and does not relate to EMH and should be removed.

Thank you for drawing our attention to this. These references have been removed.

2. Ten plant types are referred to on page 3 line 16 and again on page 8 line 25. This would be *Buchanania*, *Canarium australianum*, polydrupe *Pandanus* sp.; *Persoonia falcata* and *Terminalia* sp (from endocarps); Monocotyledonous stem-based storage organ types A and B (from vegetative parenchyma); two types of Areceaceae (stem tissue) and the tenth is seeds based on grinding stone evidence? This needs to be made clearer particularly as the article has now removed the grinding stone evidence and there appears to be no seeds found within the archaeobotanical remains. Perhaps discussing why there is a lack of seeds also needs to be considered particularly due to the grinding stone evidence.

The ten plant types refer to macrobotanical remains identified within this study. They do not include the residue and usewear evidence for seed-grinding published in Clarkson et al. (*Nature* 2017).

The ten plant types are *Buchanania* sp., *Canarium australianum*, polydrupe *Pandanus* sp. cf. *spiralis*, *Persoonia falcata*, *Terminalia* sp., **Secondary root tissue Type A**, Monocotyledonous stem-based storage organ Type A and Type B, and Areceaceae stem tissue Type A and B (see Supplementary Table 1).

We agree that it is interesting that there are no edible seeds in Phase 2, despite a methodology that should have allowed for their recovery. We are not sure why this is. It may be a question of comparatively low-intensity seed-grinding in this phase, meaning only a small opportunity for grains to be charred under the correct conditions for preservation, or it may be a factor of processing practices or locales. However, in lieu of an explanation, we have decided the best thing is to highlight this fact to the reader, so that they can consider it. The citation of this residue and usewear evidence now reads, “Furthermore, whilst there were no edible seeds recovered in the plant

macrofossil assemblage, residue and usewear evidence from the Phase 2 grinding stone assemblage has identified seed-grinding during this initial phase of occupation,” (Page 9, Lines 3-5).

3. One of the examples identified as *Curcuma australasica* rhizome in the first submission is now identified as Arecaeae stem tissue.

This is correct.

The original identification of *Curcuma australasica* was made based on *C. australasica* reference material, available literature and several very fragmented specimens from Phase 2. Both Zingiberaceae and Arecaeae families produce globular echinate phytoliths within their stem tissue. The available literature suggests that the phytoliths produced by Arecaeae are larger in diameter (>5µm) than those produced by Zingiberaceae (Benvenuto et al. 2015; Fenwick et al. 2011). Therefore, the small size of the phytoliths found in the Phase 2 specimens (2-3µm in diameter) and their general anatomical similarity to *C. australasica* stem tissue (the only Zingiberaceae endemic to northern Australia) led to this identification.

However, as both more reference material was examined, and thousands of similar, but younger and better-preserved, archaeological specimens from Madjedbebe were analysed using an SEM, it became clear that this identification was incorrect. There is more variation in the size and shape of northern Australian Arecaeae phytoliths than is suggested in the literature and the vascular bundles present in the archaeological specimens include a fibre cap, indicative of palm anatomy. Further, the parenchyma cells present in the archaeological specimens are both smaller and more thick-walled than those of *C. australasica* stem tissue.

We have, therefore, changed this identification and we hope this action is recognised as an indication of our commitment to good science.

Reviewer #3 (Remarks to the Author):

The manuscript presents the findings of some excellent archaeobotanical research at the EHM site of Madjedbebe in NW Australia. Although the age of the Phase 2 occupation presented here is questioned by some, it can be presumed to minimally predate 50K bp. As such, the materials under discussion here - especially those derived from the fills of archaeological features - are mostly associated with EHM occupation of the site.

The archaeobotanical research is very impressive and conservative in terms of the identifications, which is always welcome. I found the SOM for identifications to be excellent and convincing in terms of rationale for taxonomic identifications.

I have four minor issues that the authors may wish to consider:

1. The argument at the beginning that EMH dispersal is more associated with faunal exploitation than floral exploitation does not really ring true. Despite the paucity of sites, and the lack of robust archaeobotanical recovery at most,

some of the oldest EMH sites in Sunda and Sahul are clearly associated with plant exploitation, including of toxic foods. Maybe start the argument with a different emphasis - these new findings at Madjedbebe reinforce the findings from Niah and Ivane Valley - and present robust evidence for the exploitation of a more diverse plant assemblage than at any other early site. I would take this to be the key point.

Similarly, suggested broadening of diet in Holocene and association with agriculture is a very old idea - the reference used is Flannery 1969 - actually agriculture likely leads to restriction of diet in many places. So, I think this is a false start to get the argument going - it would be much better to start with early evidence from MSA sites in Africa, Niah and Ivane - and go from there ...

We agree with the arguments raised and think that the evidence currently available for the use of plants in Sunda, Wallacea and Sahul (ie. Niah Cave and Ivane Valley) does suggest that early modern human populations used a range of plant foods, some of which required intensive processing to make them edible.

However, this is not how the available evidence has been interpreted by those who champion the single coastal dispersal model (Mellars et al. 2013, O'Connell and Allen 2012). For example, O'Connell and Allen (2012) interpret the use of toxic plant species at Niah Cave, not as representative of the earliest diet in this region, but as evidence of a "decline in the availability of high ranked resources" within Sunda, explaining the push factor which drove populations to continue through to exploit high-ranked patches available in Wallacea and Sahul. For this reason, we think it is important to highlight this debate and the implications new understandings of early plant use may have on it.

However, as suggested, we have shifted the emphasis. We have argued that: "Current evidence for Pleistocene plant use in Sunda and Sahul, whilst not necessarily related to the earliest phases of human expansion in this region, largely supports this latter interpretation. This is because the intensive and multi-step processing techniques required to make the identified plant foods edible are indicative of both complex and flexible foraging, and the kind of broad diet that underpins adaptations to more difficult environments," (Page 2, Lines 26-32).

We also agree that Flannery's 'Broad Spectrum Revolution' is an older idea. However, we believe it is still relevant in discussions today and we have added Stiner's (2001) more recent and well-cited review of this theory to the references to demonstrate this. We have also rewritten this sentence to make it clear that this was not an argument for a broadening of diet with the origins of agriculture, but for a broadening of diet in the millennia prior: "Extensive use and processing of plant resources, and an associated broadening of the diet, was therefore typically considered a late Pleistocene/early Holocene phenomenon, linked to changing foraging behaviours in the millennia prior to the emergence of agriculture," (Page 2, Lines 1-4). This broadening of diet is still argued by some to have led to the foraging of the wild ancestors of crop

products, their consequential domestication and an eventual restriction of diet with agricultural practices.

We have also further nuanced the introductory remarks, making it clear that there has been a more recent paradigm shift towards an understanding of the importance of plant foods in early modern human diet. For instance, we have written, "More recent research into plant macro- and micro-fossils is breaking down this paradigm," (Page 2, Lines 5-6), rather than "beginning to break down this paradigm." And, we have also added that, "This shift in paradigm is particularly important when considering the southern dispersal of *Homo sapiens* out of Africa," (Page 2, Lines 16-17).

2. Although the plants identified require processing and cooking - these are not as necessarily labour intensive as suggested in the manuscript. The archaeobotany of EMH occupation at Niah suggests toxic plant food processing, whereas this is not necessarily implied by the Madjedbebe data. So, perhaps tone this aspect down.

Thank you. We have toned down the statements around this concept, so they now read, "plant remains, including those requiring processing," (Page 1, Lines 35-36; Page 2, Lines 44-45), rather than those "requiring labour-intensive processing".

However, we still mention that some of the plants present in the Phase 2 assemblage likely required multi-step processing techniques, which comprised pounding, cooking and grinding. We have, therefore, added a clarifying sentence to the discussion: "Whilst none of these plants are toxic, the processing required to extract and make edible the nutritious components from some of the taxa present is suggestive of multi-step and labour-intensive processing techniques," (Page 9, Lines 5-7).

3. Pandanus extraction experiment is completely out-of-place and nonsensical - no doubt based on urban dwellers who have no experience with extracting materials, other than cracking the odd walnut or peeling a banana. There is no comparison - as a skilled practitioner could probably do work with Pandanus as easily as we would do these mundane activities with walnuts and bananas. I would suggest omitting the '54 blows' sentence and associated SOM. Just think - how long would it take most people to milk a cow relative to someone who knows what they are doing?

We agree that there is a disconnect between our amateur urban-dwelling experimental participants and the early inhabitants of Madjedbebe. As suggested, we have both removed this sentence and the associated SOM.

4. Antiquity of plant remains is a thorny issue. As I understand it, most of these materials cannot be directly radiocarbon dated because they are at/beyond the range of the technique. Ok. However, that leaves the issue of archaeological association and antiquity unresolved for those archaeobotanical fragments distributed through the stratigraphy, rather than those collected from the hearth (which are presumably more secure in terms

of chronostratigraphic association). Surely a more relevant SOM than the Pandanus extraction experiment would be to look at the age-depth comparison for the site as a whole between radiocarbon and OSL dates - showing how the ability to date carbon-based materials (such as older plant fragments) tails off downward. I also wonder if there is carbon replacement in some archaeobotanical remains - especially more porous parenchyma - which could actually be measured directly in those fragments to show that there is increasing replacement of carbon with depth; hence, meaning they cannot be directly dated (even if not beyond the radiocarbon limit). Just a thought.

Thank you for your constructive considerations around this issue. We have adopted your suggestion and added a figure describing the age-depth comparison between OSL and AMS radiocarbon dates to our SOM. The text now reads, "Its earliest, dense phase of occupation (Phase 2) contains charcoal, abundant ground ochre, grinding stones, including those used for seed-grinding, and a dense assemblage of unique flaked stone artefact types and raw materials (>10,000 artefacts). This phase is dated to c.65-53kya on the basis of an extensive single-grain optically stimulated luminescence and AMS radiocarbon dating regime (Supplementary Fig. 1)," (Page, Lines).

In addition, the text also now cites Clarkson et al. (2018), which considers the issue of carbon replacement with depth at Madjedbebe by using the dissolution of charcoal samples during the base phases of ABA and A-BOX pre-treatment as a proxy for oxidative degradation of the original carbon (Ascough et al. 2011).

5. Lastly - use of 'bush foods' in title is somewhat misleading - as only has currency in Australian context - even if it will play well in the media.

We agree that the term 'bush foods' may not be prevalent outside Australia and have change the title to read, 'The First Australian Plant Foods at Madjedbebe, 65,000–53,000 years ago'.

References:

Ascough, P. L., Bird, M. I., Francis, S. M. & Lebl, T. Alkali extraction of archaeological and geological charcoal: evidence for diagenetic degradation and formation of humic acids. *Journal of Archaeological Science* **38**, 69-78, doi:10.1016/j.jas.2010.08.011 (2011).

Benvenuto, M. L., FernÁNdez Honaine, M., Osterrieth, M. L. & Morel, E. Differentiation of globular phytoliths in Arecaceae and other monocotyledons: morphological description for paleobotanical application. *Turkish Journal of Botany* **39**, 341-353, doi:10.3906/bot-1312-72 (2015).

Clarkson, C. *et al.* Reply to comments on Clarkson et al. (2017) 'Human occupation of northern Australia by 65 000 years ago'. *Australian Archaeology* **84**, 84-89, doi:10.1080/03122417.2018.1462884 (2018).

Fenwick, R. S. H., Lentfer, C. J. & Weisler, M. I. Palm reading: a pilot study to discriminate phytoliths of four *Arecaceae* (*Palmae*) taxa. *Journal of Archaeological Science* **38**, 2190-2199, doi:10.1016/j.jas.2011.03.016 (2011).

Northern Territory Government, *Flora NT: Northern Territory flora online*, <<http://eflora.nt.gov.au/home>> (2013).

O'Connell, J. F. & Allen, J. The restaurant at the end of the universe: Modelling the colonisation of Sahul. *Australian Archaeology* **74**, 5-17, doi:doi:10.1080/03122417.2012.11681932 (2012).

Stiner, M. C. Thirty years on the "Broad Spectrum Revolution" and paleolithic demography. *Proceedings of the National Academy of Sciences* **98**, 6993-6996, doi:10.1073/pnas.121176198 (2001).